# The Usefulness of Magnetic Resonance Imaging in the Management of Acute Phlegmonous Esophagitis

**DOI:** 10.3390/diagnostics13152600

**Published:** 2023-08-04

**Authors:** Motohiro Shimizu, Hiroshi Adachi, Kenichi Sai

**Affiliations:** Department of Intensive Care Medicine, Ryokusen-kai Yonemori Hospital, Kagoshima 890-0062, Japan

**Keywords:** acute phlegmonous esophagitis, abscess, magnetic resonance imaging

## Abstract

Acute phlegmonous esophagitis is a rare life-threatening disease that often requires surgical intervention in case of complications, including esophageal abscess, perforation, or mediastinitis. We present a case of acute phlegmonous esophagitis, in which magnetic resonance imaging (MRI) proved useful in planning the treatment strategy. An 89-year-old woman was admitted to the emergency department with painful swallowing and respiratory distress. She was diagnosed with acute phlegmonous esophagitis and a hypopharyngeal abscess based on computed tomography (CT) findings. However, there was a discrepancy between the clinical course and CT findings. Given the improvement of the patient’s condition with conservative treatment with ampicillin/sulbactam, the CT findings suggested an apparent abscess due to increased esophageal wall thickness. However, MR diffusion-weighted images showed a slightly high-intensity signal, suggesting that the enlargement was due to edema rather than an abscess. The patient recovered successfully following conservative treatment. Thus, our findings demonstrate the utility of MRI in the treatment planning of acute phlegmonous esophagitis, especially in cases with unreliable contrast-enhanced CT findings. However, future studies are warranted to explore the utility of MRI in the management of such cases.


Figure 1Contrast-enhancd CT of the neck and chest on the day of admission. An 89-year-old woman who experienced painful swallowing for 4 days suddenly developed severe dyspnea and was admitted to our emergency department. Upon arrival at the hospital, she was febrile (38.0 °C) with a Glasgow Coma Scale score of 10 (E3V1M6), blood pressure of 154/114 mmHg, heart rate of 140 bpm, and respiratory rate of 40 breaths per minute. Physical examination indicated a diagnosis of stridor. The patient was immediately intubated due to upper airway obstruction. The arterial blood gas analysis revealed respiratory acidosis (pH, 7.309; PaCO_2_, 56 mmHg), and laboratory findings revealed elevated white blood cell (21,800/μL) and C-reactive protein (15.8 mg/dL) levels. Contrast-enhanced computed tomography (CT) imaging of the neck and chest revealed a low-density area in the right hypopharynx ((**A**): arrow) as well as diffuse thickening of the esophageal wall with an intramural circumferential low-density area ((**B**,**C**): arrows). Potential differential diagnoses suggested by the CT findings, including corrosive esophagitis, reflux esophagitis, dissecting intramural hematoma, and esophageal duplication [1], were ruled out, taking into account the patient’s medical history, physical examination, and laboratory assessments. Based on these findings, the patient was diagnosed with acute phlegmonous esophagitis with a hypopharyngeal abscess and was administered empirical antibiotic treatment with intravenous ampicillin/sulbactam (3 g every 6 h). Blood cultures performed on admission revealed the presence of *Streptococcus constellatus*, sensitive to ampicillin/sulbactam.
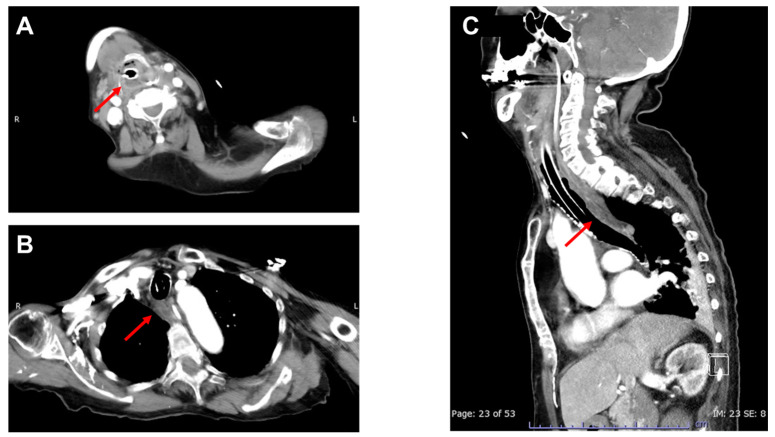

Figure 2Contrast-enhanced CT of the neck and chest on the fourth day after admission. On the fourth day after admission, the patient’s hemodynamics improved, and body temperature decreased below 37.5 °C; moreover, there was a decrease in inflammatory markers (white blood cells: 13,900/μL; C-reactive protein: 10.9 mg/dL). Contrastingly, enhanced CT revealed an enlargement of the esophageal wall thickening with low intramural circumferential density and a peripheral enhancement rim ((**B**,**C**): arrows), which were suggestive of an abscess in the esophageal wall (Figure 2). Meanwhile, the low-density area in the right hypopharynx showed no significant change((**A**): arrow). Acute phlegmonous esophagitis was mainly diagnosed using CT [1,2,3]. However, in our case, even though the diagnosis was established based on CT findings, the contrast-enhanced CT findings were inconsistent with the clinical course of the patient. Endoscopy performed on the same day revealed edematous mucosa in the esophagus; however, there was no pus or perforation (Figure 3). Additionally, there were no findings suggestive of corrosives or reflux esophagitis.
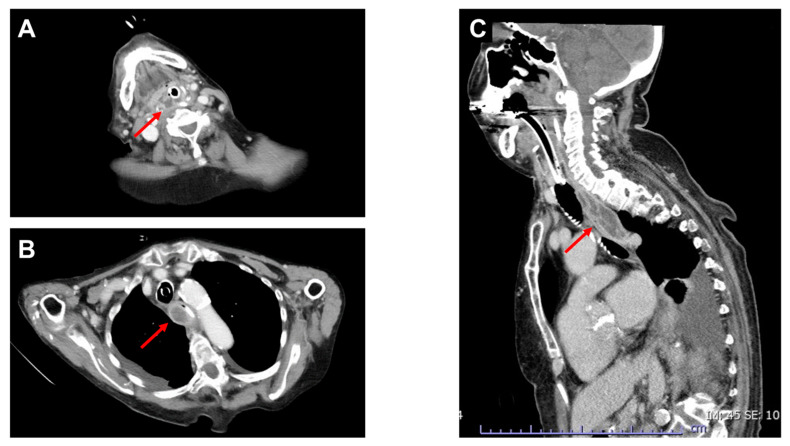

Figure 3Endoscopic findings of the upper esophagus on the fourth day after admission.
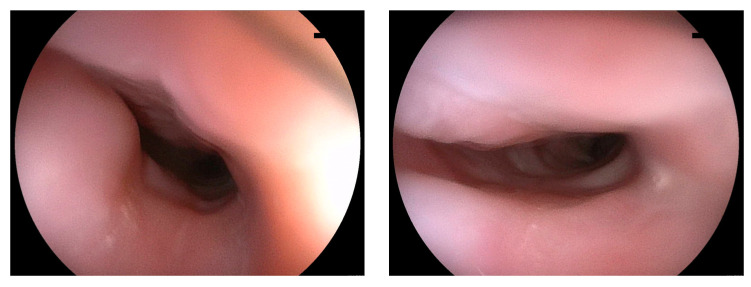

Figure 4MRI of the neck and chest on the fourth day after admission ((**A**–**C**): T2-weighted images; (**D**–**F**): diffusion-weighted images; (**G**–**I**): apparent diffusion coefficient maps). The patient then underwent an MRI. T2-weighted images showed thickening of the hypopharyngeal ((**A**): arrow) and esophageal walls ((**B**,C): arrows). On diffusion-weighted images, the hypopharyngeal wall thickening showed a high-intensity signal ((**D**): arrow), whereas the esophageal wall thickening showed a slightly high-intensity signal ((**E**,**F**): arrows). Similarly, on apparent diffusion coefficient maps, the hypopharyngeal and esophageal wall thickening showed a low ((**G**): arrow) and slightly low-intensity signal ((**H**,**I**): arrows), respectively. The MRI findings indicated that the enlargement of the esophageal wall thickening observed on follow-up enhanced CT was due to edema rather than an abscess; accordingly, the conservative treatment was continued. Although the patient’s condition improved, she underwent a tracheostomy on the eighth day of hospitalization for laryngeal edema caused by a hypopharyngeal abscess. After 5 weeks, the tracheostomy cannula was removed as the patient’s condition improved. The patient was successfully discharged on the 70th day after admission. Acute phlegmonous esophagitis is a rare condition, with only 20 published case reports in English, including ours. To our knowledge, this is the first report to demonstrate the utility of MRI in managing acute phlegmonous esophagitis. In our case, MRI was used to rule out the presence of an abscess in the esophageal wall, with the patient’s condition improving after conservative treatment. Thus, in case of difficulty differentiating between edema and abscesses due to increased esophageal wall thickening on enhanced CT, MRI can offer additional insights, which might contribute to the decision-making process on whether surgical intervention is required in managing acute phlegmonous esophagitis. MRI is useful in assessing the severity of neck infections [4]; furthermore, it is more accurate than CT in diagnosing abscesses [5]. Moreover, in conditions such as deep neck infections [6] and orbital cellulitis [7], MRI has proven helpful in distinguishing between abscesses and edema. Specifically, T2-weighted images can be used to assess normal anatomy and tissue edema, whereas diffusion-weighted images can effectively evaluate abscess formation. Similarly, MRI has proven beneficial in diagnosing abscesses in musculoskeletal soft-tissue infections [8]. Our findings suggest that MRI is effective in detecting apparent abscesses, even in patients with acute phlegmonous esophagitis. In contrast, two previous studies demonstrated that endoscopic ultrasonography (EUS) is useful for diagnosing abscesses in the esophageal wall [1,9]. However, EUS requires a well-trained physician and a dedicated endoscope, which are currently unavailable at our facility. In conclusion, our case highlights the utility of MRI in planning the treatment of acute phlegmonous esophagitis, especially in cases with unreliable contrast-enhanced CT findings. Specifically, MRI allows for the precise assessment of esophageal wall abscesses, which could improve patient outcomes through appropriate conservative treatment. Future studies are warranted to further explore the usefulness of MRI in the management of acute phlegmonous esophagitis and related complications.
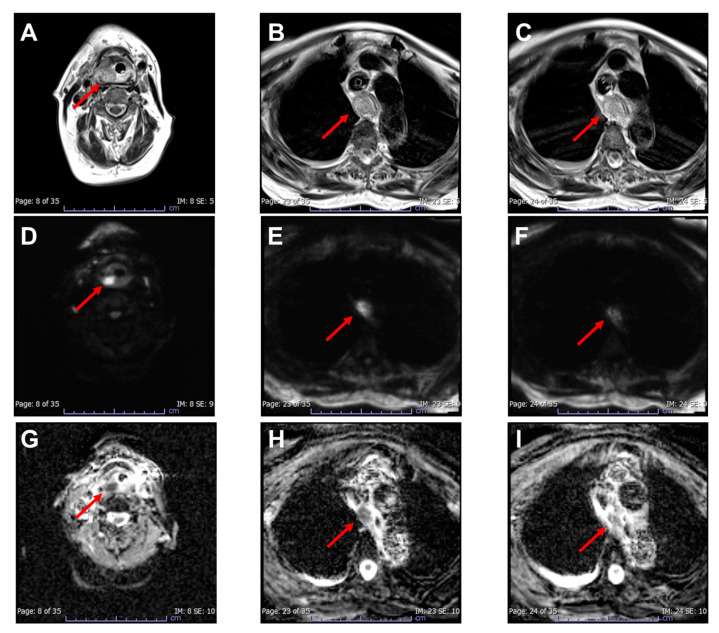



## Data Availability

Not applicable.

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
