# Peer review of "The Usefulness of Magnetic Resonance Imaging in the Management of Acute Phlegmonous Esophagitis"

_diagnostics, 2023, doi:10.3390/diagnostics13152600_

Round 1

Reviewer 1 Report

Thank you for the privilege of reviewing your work. This manuscript is well written. I think this report interesting.

The authors mainly described the efficacy of MRI in the treatment planning of acute phlegmonous esophagitis.

1.      Please described the reports of other diseases regarding the difference edema and abscesses in MRI.

2.      Please add the figure of Endoscopy.

3.      Please move L56-57 to the right place.

English is well written

Author Response

Thank you for reviewing my manuscript. We basically agreed with your comments and revised accordingly.

The major parts of revisions are as follows:

  1. We have added the descriptions regarding conditions in which MRI is useful in distinguishing between abscesses and edema, as well as diagnosing abscesses.
  2. We have added the figure of endoscopy images.

Point 1: Please described the reports of other diseases regarding the difference edema and abscesses in MRI.

Response 1: Following your comments, we have cited deep neck infections and orbital cellulitis as conditions where MRI is useful in distinguishing between edema and abscesses. Furthermore, we have noted that MRI is beneficial for diagnosing abscesses even in musculoskeletal soft-tissue infections.

Point 2: Please add the figure of Endoscopy.

Response 2: As suggested, we have added the figure of endoscopy images (Figure 3).

Point 3: Please move L56-57 to the right place.

Response 3: In response to your comments, we have moved the following sentence: “Acute phlegmonous esophagitis is a rare condition, with only 20 published case reports in English, including ours.”

Reviewer 2 Report

This case report demonstrated the utility of MRI in the accurate diagnosis and treatment planning of acute phlegmonous esophagitis, especially in case of unreliable contrast-enhanced CT findings.

This report has a certain reference significance for such clinical cases.

The manuscript was well written.

Author Response

Thank you for reviewing my manuscript. We revised accordingly.

The major parts of revisions are as follows:

  1. We have added the descriptions regarding conditions in which MRI is useful in distinguishing between abscesses and edema, as well as diagnosing abscesses.
  2. We have added the figure of endoscopy images.